# Emotional Intelligence among Nursing Students: Findings from a Longitudinal Study

**DOI:** 10.3390/healthcare10102032

**Published:** 2022-10-14

**Authors:** Leona Cilar Budler, Lucija Gosak, Dominika Vrbnjak, Majda Pajnkihar, Gregor Štiglic

**Affiliations:** 1Faculty of Health Sciences, University of Maribor, Žitna Ulica 15, 2000 Maribor, Slovenia; 2Usher Institute, University of Edinburgh, Edinburgh EH8 9YL, UK

**Keywords:** emotional intelligence, nursing, students, caring experience, TEIQue-SF, SSEIT

## Abstract

Emotional intelligence is an important factor for nursing students’ success and work performance. Although the level of emotional intelligence increases with age and tends to be higher in women, results of different studies on emotional intelligence in nursing students vary regarding age, study year, and gender. A longitudinal study was conducted in 2016 and 2019 among undergraduate nursing students to explore whether emotional intelligence changes over time. A total of 111 undergraduate nursing students participated in the study in the first year of their study, and 101 in the third year. Data were collected using the Trait Emotional Intelligence Questionnaire Short Form (TEIQue-SF) and Schutte Self Report Emotional Intelligence Test (SSEIT). There was a significant difference in emotional intelligence between students in their first (M = 154.40; 95% CI: 101.85–193.05) and third year (M = 162.01; 95% CI: 118.65–196.00) of study using TEIQue-SF questionnaire. There was a weak correlation (r = 0.170) between emotional intelligence and age measuring using the TEIQue-SF questionnaire, and no significant correlation when measured using SSEIT (r = 0.34). We found that nursing students’ emotional intelligence changes over time with years of education and age, suggesting that emotional intelligence skills can be improved. Further research is needed to determine the gendered nature of emotional intelligence in nursing students.

## 1. Introduction

Until recently, in many environments, the golden standard for student’s success was the intelligence quotient (IQ), which was also frequently used for entry tests in higher education. However, among these intelligent people, many do not face stressful situations and do not socialise with people. The consequence is that many of them fail in personal and professional life. Therefore, emotional intelligence (EI) was proposed as one of the measures that envision an individual’s all-around development and success [1,2].

EI reflects the capacity to comprehend and regulate emotions and cope effectively with emotional situations [3]. It can also be described as the ability, skill, or self-perceived ability to recognise, evaluate, and manage the emotions of ourselves, others, and groups [4]. Schmidt and Hunter [5] defined EI as the ability to grasp and reason correctly with concepts and solve problems. Petrides and Furnham [6] state that EI consists of two distinct concepts: an emotion-related cognitive ability and behavioural dispositions and self-perceptions of one’s ability to recognise and understand emotions. Intellect and EI work separately, as the individual may be intellectually brilliant but emotionally incapable. The IQ contributes only 20% to lifetime success, the rest is the result of EI, including motivation, perseverance, impulse control, empathy, and hope [7].

EI is now widely used among different organisations and schools [8,9] as it may contribute to the better performance of individuals both personally and professionally [9,10]. The nursing profession includes considerable emotional work, including managing and expressing relevant emotions. Therefore, researchers and lecturers need to focus on empowering EI in nursing to improve the nursing profession [1]. Nurses with a high EI level are more empathic, compassionate, caring, and resilient [11].

EI may be an important factor for nursing students’ success [10,12,13] and retention [14]. It can allow nursing students to face challenges in clinical placements [14] effectively, improve their leadership skills, performance in practice, and to enhance patient safety [15]. Several studies found that EI levels increases with age [10,16] and among nursing students in their first and last year [15,16,17]. EI also tends to be higher in women [18]; however, results of different studies on EI in nursing vary regarding age, study year, and gender [14]. Foster et al. [15] and Nwabuebo [19], for example, did not find age or gender differences in EI scores. Hajibabaee et al. [18] found that students in their first and third years had the highest and lowest emotional intelligence scores, respectively. Foster et al. [15], on the other hand, found a significant increase in students EI at the beginning of the second year; however, there was not a significant change at the end of the study program, when compared with their study entry.

To the best of our knowledge, few longitudinal studies were conducted on EI in nursing students [13,15,20,21], and none from the Slovenian perspective. Therefore, the current study aimed to determine EI among nursing students and explore whether EI in nursing students changes over time in a Slovenian university. Objectives were to establish whether there were statistically significant differences in EI scores in nursing students in their first study year compared with the third study and to determine differences between EI scores and nursing students’ age.

## 2. Materials and Methods

### 2.1. Study Design

A longitudinal study was carried measuring EI scores using self-reported measures at two time points over three years.

For the purposes of the longitudinal study, a register of codes or identification numbers and the names and surnames of the respondents was set up due to the nature of the study and the data that needed to be collected on academic performance. The codes and names of the respondents were stored separately from the completed questionnaires and the academic performance data, in locked rooms, and were not accessible to higher education teachers and staff who were part of the study process. The data were collected and stored in a database which was properly password-protected and the data were encrypted and anonymised. Certain codes were used by the students throughout the longitudinal study.

The data were anonymised; this method of data collection was approved by the Ethics Committee of the University of Maribor, Faculty of Health Sciences (No. 038/2016/5975-2/1/504).

### 2.2. Participants

The study was conducted among undergraduate nursing students at the nursing faculty in Slovenia. The survey was conducted between 16th November 2016 and 30th January 2017, and between 12th November 2018 and 31st January 2019. The questionnaires were administrated during the seminars and laboratory exercises. The sample consisted of 111 undergraduate nursing students who were enrolled in a nursing study program in the 2016/2017 academic year. The same sample of students was invited to fill the questionnaire in 2019 in their third (i.e., last) study year: 101 students filled out the questionnaire; however, some students failed to write their ID numbers appropriately. As such, only 77 questionnaires were matched longitudinally.

### 2.3. Measures

All participants completed the questionnaire with EI measures on the entry in year one (2016) and at the end of year three (2019). This study is a replication of the Snowden et al. (2015); therefore, EI was measured using the Trait Emotional Intelligence Questionnaire Short Form (TEIQue-SF) [6,22,23] and Schutte Self Report Emotional Intelligence Test (SSEIT) [24].

The TEIQue-SF is a 30-item scale with responses on a 7-point Likert-type scale ranging from 1 (strongly disagree) to 7 (strongly agree). Four subscales are included: emotionality, sociability, self-control, and well-being [25,26]. Of 30 items, 15 are negatively stated and need reverse-scoring [6]. The total scale scoring is derived by summing the score on each item in the scale. The higher the score, the higher the trait EI of the individual [19,22]. The SSEIT is a 33-item scale with four subscales: emotion perception, utilising emotions, managing self-relevant emotions, and managing others’ emotions. The scale ranges from 1 (strongly agree) to 5 (strongly disagree). Each subscale score is graded and then summed to obtain the total score [24]. Both measures were back-translated into the Slovene language [10].

### 2.4. Data Analysis

Data were analysed using IBM SPSS Statistics (IBM, version 25, Slovenia) and R (3.6.1) and tested for normality and homogeneity of variance. Correspondingly, parametric, or non-parametric statistical tests were used to assessing the impact of different variables on SSEIT and TEIQ scores in the first and third study year. Due to the longitudinal study design, we were able to use paired-samples *t*-test to test for differences based on the study year. Spearman correlation coefficient was used to measure the correlation between age and two EI measures.

## 3. Results

### 3.1. Demographics

In total, 77 students were included in the study. Students’ average age was 19.27 ± 0.81, with a minimum age of 18 and a maximum age of 23 years. In the first study year, 66 (85.7%) of students attended full-time study programs and 68 (88.3%) in the third study year.

In total, 55 (71.4%) students already had practical experience in healthcare. Almost half of them (n = 38; 49.4%) acquired their hospital experience during their practical education. Most students had previously completed secondary school for nursing (n = 52; 67.5%). Other students completed general secondary school (n = 18; 23.4%), technical secondary school (n = 3; 3.9%) or other schools (n = 4; 5.2%). When the study was conducted in the first study year, two (2.6%) students were employed, and eight (10.4%) students were employed in the third study year (Table 1).

Table 2 shows the differences between the SSEIT Score and the TEIQ Score in the first and third year of study according to gender, dominant hand, the environment in which the person grows up, employment status, the relationship status, the birth order, the number of hours spent at work each month, and work performance.

The uneven distribution of students across demographics makes it difficult to speak of significant differences. Students who were not employed had a higher TEIQ Score (155.21; SD = 23.21) in the first study year. In the third year, the highest score was found for the middle-born child (167.74; SD = 17.99). In the first study year, students who spent less than 10 h per month at work had higher SSEIT (94.16 vs. 82.75) and TEIQ (154.79 vs. 138.75) scores. In the third study year, the results were the opposite, with higher scores for those who spent more time at work for SSEIT score (103.50 vs. 95.06) and for TEIQ score (194.00 vs. 160.55). Students who rate their work performance as highly successful have a higher SSEIT Score (98.30 vs. 90.18 for the first study year; and 96.38 vs. 94.29 for the third study year) and TEIQ score (167.78 vs. 146.52 for the first study year; and 158.00 vs. 165.08 for the third study year) compared with those who rate their work performance as average.

### 3.2. EI Scores between Nursing Students in the First and Third Year of Study

Based on the Shapiro–Wilk test, we found that mean TEIQue-SF scores between the nursing students in the first study year and nursing students in the third study year were normally distributed. Figure 1 shows mean TEIQue-SF scores for nursing students in the first study year with a mean value of 154.40 (95% CI: 101.85–193.05) and for nursing students in the third study year with a mean of 162.01 (95% CI: 118.65–196.00).

We used paired-samples T-test and found a statistically significant difference in TEIQue-SF scores between nursing students in the first study year and nursing students in the third study year (t (76) = −3.390; *p* = 0.001).

According to the Shapiro–Wilk test, the distribution of mean SSEIT scores between the nursing students in the first and third study year was normal. The mean SSEIT scores for nursing students in the first study year were 93.26 (95% CI: 67.90–118.20) and for nursing students in the third study year 95.34 (95% CI: 71.95–124.15).

Based on the paired-samples T-test, we found no statistically significant difference in SSEIT scores between nursing students in the first study year and nursing students in the third study year (t (76) = −1.523; *p* = 0.132). Figure 2 shows the distribution of the scores as well as the mean and 95% confidence interval values for the SSEIT score.

## 4. Discussion

This longitudinal study was conducted to investigate whether EI in nursing students changes over time. We confirmed a higher EI of students in the third year of study compared with students in the first study year. In our study, the difference was statistically significant, measuring the level of EI using TEIQue-SF, but not statistically significant when measured using SSEIT. Similar to our findings, Benson et al. [17] found that the average total EI was 98.0 for first-year students, 103.72 for the second year, 104.56 for the third year, and 107.80 for the fourth year, which shows us that EI has grown with study year. Herr et al. [27] also reported that third-year nursing students (M = 125.39; SD = 8.71) had higher EI than first-year students (M = 124.69, SD = 11.76), when measuring EI with SSEIT scale, but results were not statistically different (t = 0.257, *p* = 0.798). Other studies found same results [13] and suggest it is possible the difference is due to the type of EI being measured. TEIQue-SF is designed to measure EI as a trait, whereas SSEIT sees EI more as an ‘ability’, in other words, more stable over time [13]. The findings could suggest that emotional skills can be improved. Students may also grow and mature over the years of education, but other factors outside of the study program may impact the findings [15]. Vernon et al. (2008) and many other heritability studies of personality found EI largely attributable to genetic and nonshared environmental factors.

This study also aimed to determine whether EI scores of nursing students differ among students’ age. Age is a significant factor affecting an individual’s emotional maturity, which Por and others also found in their study who identified a strong positive relationship between the age of nursing students and their EI [28]. Ishii [29] also found out that EI positively correlates with age. EI evolves through life as we encounter different experiences, and as a result, our competence grows [30,31,32,33]. EI in this study was significantly correlated with age when TEIQue-SF was used but not when measured using SSEIT. It should be noted that Slovenia students usually study nursing at a young age compared with some other countries. The narrow age range of just five years (from 18 to 23) confirms this.

It is known that females have a higher level of EI than males [9,13,29,32,33]. In males, EI is often correlated with the inability to perceive emotions, negative behaviours [34], and poor interpersonal relationships [32]. Due to the unequal gender distribution of the students, it was not possible to assess the differences on SSEIT scores and the TEIQ scores. Thus, further research is warranted with an equal number of male and female nursing students. We found some significant differences between the demographics and SSEIT scores and the TEIQ scores, but this is not significant due to the uneven distribution of the population between the groups.

Priyam et al. [35] found that people who have higher emotional intelligence score are more successful, accurate and precise in their work than those with lower emotional intelligence score. Similarly, our study confirms that people who believe they are high performers at work have higher SSEIT and TEIQ scores in their first and third year. Statistical correlation was found only in the first year of study.

The results should be therefore interpreted with caution, and some other limitations need to be considered: only one Slovenian nursing faculty was included; most of the participants were full-time students; there was also a loss during follow-up over the study, and self-reported measures were used.

Despite limitations, our study adds to the body of knowledge on EI in nursing students, especially in the Slovenian context. Findings suggest that EI skills can be improved; therefore, more attention should be given to EI skills development, and EI should be implemented into the nursing curriculum.

Additionally, further studies should use performance-based measures of EI and not only instruments that report students’ own level of EI [36].

## 5. Conclusions

Nursing students interact daily with people from different cultures and backgrounds; thus, they must develop different skills and knowledge to ensure a high quality of healthcare. Although it is known that women have a higher EI level, our research in 2018 in Slovenia did not confirm this, and further research is warranted. Moreover, the EI level usually increases with age, meaning that students in the third year should have a higher EI level than students in the first year. This was also confirmed in our study. It would be interesting to follow up with students after transitioning from being a student to a registered nurse working in practice to determine EI changes over their years of experiences in practice and professional development and investigate whether workshops and training courses on EI influence EI their EI scores after graduation.

## Figures and Tables

**Figure 1 healthcare-10-02032-f001:**
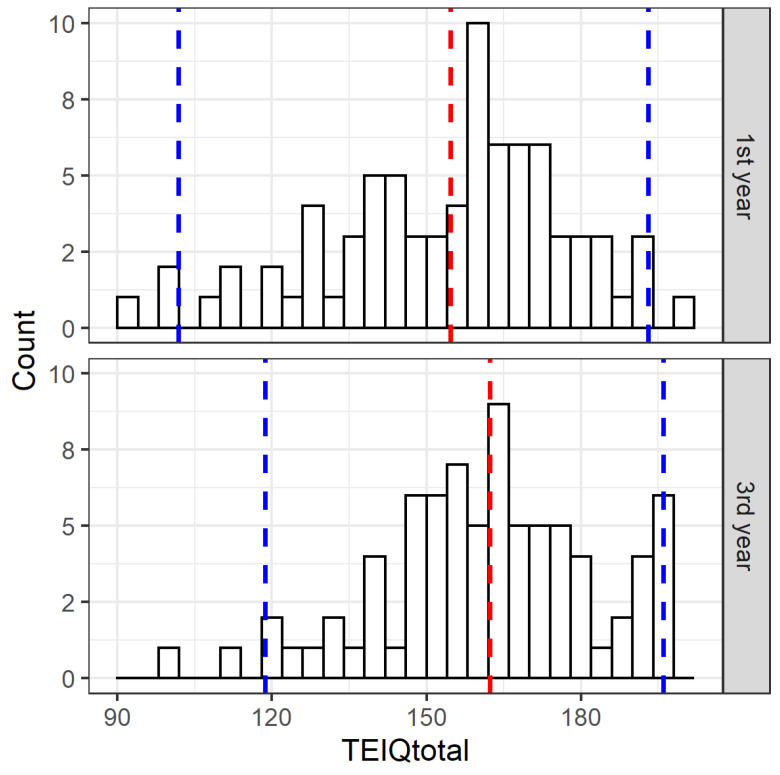
Distribution of TEIQue-SF scores by year of study with mean value (blue) and confidence interval (red) in the first year and third year of study.

**Figure 2 healthcare-10-02032-f002:**
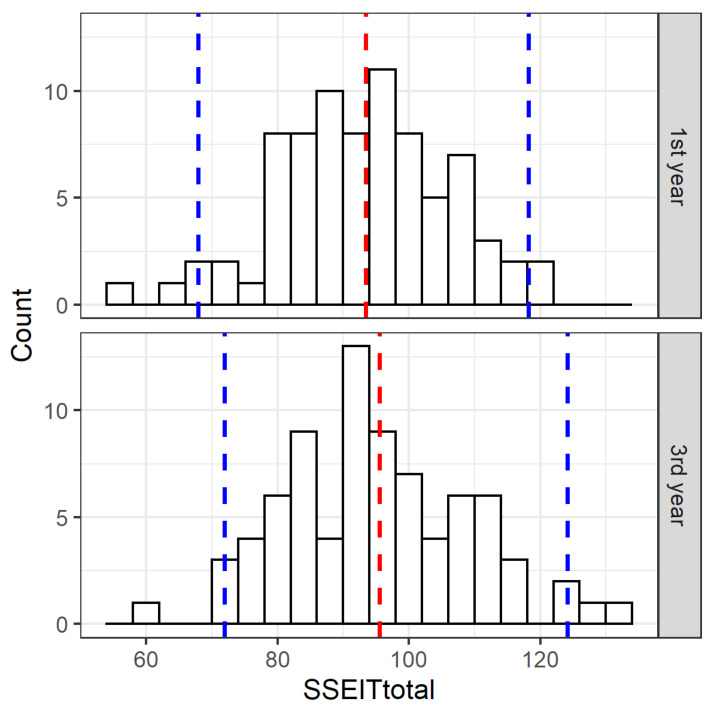
Distribution of SSEIT scores by year of study with mean value (blue) and confidence interval (red) in the first year and third year of study.

**Table 1 healthcare-10-02032-t001:** Distribution of students by gender, year of study, and study program.

		First Study Year		Third Study Year	
		Full-Time	Part-Time	Full-Time	Part-Time
Gender	Male	4	2	4	2
	Female	62	9	64	7
Total		66	11	68	9

**Table 2 healthcare-10-02032-t002:** Average SSEIT and TEIQ scores in the first and third year in relation to other variables.

		First Study Year						Third Study Year					
		SSEIT Score			TEIQ Score			SSEIT Score			TEIQ Score		
		Mean	SD	*p*-Value	Mean	SD	*p*-Value	Mean	SD	*p*-Value	Mean	SD	*p*-Value
Dominant hand	Left	92.40	14.58	0.410	154.70	34.14	0.488	94.10	12.45	0.381	156.70	26.00	0.198
	Right	93.39	12.50		154.36	21.77		95.52	14.02		162.81	20.29	
Environment	Town	95.53	9.52	0.359	156.47	29.86	0.632	95.76	13.05	0.895	167.76	26.47	2.985
	Village	93.22	13.51		153.21	22.45		95.47	14.21		160.29	19.45	
	Other	85.25	8.85		164.00	4.69		88.00	5.66		163.00	7.07	
Employed	Yes	91.00	4.24	0.400	124.00	4.24	0.031	90.88	12.69	0.247	171.88	21.26	0.488
	No	93.32	12.84		155.21	23.21		95.86	13.87		160.87	20.84	
Marital status	Single	93.40	10.62	0.239	153.74	22.83	0.626	94.42	13.57	0.786	161.87	20.41	0.729
	Living with partner	95.85	16.14		159.35	24.41		96.46	15.77		160.50	22.83	
	Other	90.64	12.35		148.86	24.97		97.13	8.24		167.38	20.57	
Sibling birth order	Firstborn	93.52	13.19	0.437	152.69	23.73	0.276	91.83	14.19	0.222	154.62	21.87	0.038
	Middle child	94.32	12.72		158.66	20.84		97.42	14.04		167.74	17.99	
	Last born	88.50	11.07		143.20	29.67		97.60	9.89		161.70	24.45	
Number of hours you spend at work each month	<10 h	94.16	12.69	0.170	154.79	22.74	0.325	95.06	13.76	0.692	160.55	20.77	0.051
	10–20 h	89.40	8.85		161.60	21.84		96.25	17.73		172.00	15.58	
	>20 h	82.75	13.60		138.75	36.50		103.50	6.36		194.00	5.66	
Performance at work	Average performers	90.18	13.69	0.004	146.52	21.99	<0.001	94.29	15.14	0.258	158.00	21.04	0.073
	Highly performers	98.30	9.66		167.78	21.08		96.38	12.40		165.08	20.63	

## Data Availability

Data are available upon request from the corresponding author.

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
