# Peer review of "Emotional Intelligence among Nursing Students: Findings from a Longitudinal Study"

_healthcare, 2022, doi:10.3390/healthcare10102032_

Round 1

Reviewer 1 Report

The present study aimed at examining whether EI in nursing students changes over time (from the first to the last year of their education). Moreover, sex and age differences in the two applied self-report EI measures were assessed. After having read this MS carefully I have to make several critical comments outlined in the following.

To begin with, it is at best a bad joke to examine sex differences (for a high-impact journal such as Healthcare) by using a sample of “6” male students. That is an absolute nonsense and has to be cancelled.

Although the authors only found a significant difference in self-estimated EI scores between the first and the last year of education for the TEIQue-SF but not for the SSEIT, the authors argued/concluded that these findings indicate that EI skills can be learned (without discussing/interpreting the different findings for the two EI questionnaire measures with any single word). Second and even more important, it remains unclear whether the EI differences are due to differences in age, nursing education, or social desirability and so on. From my point of view the authors should also have administered a performance-based measures of emotional abilities/skills instead of using two self-estimate questionnaires (by the way, the authors did not provide a rational for using two EI questionnaires). From my point of view, it is not convincing to assess whether students think/believe that they have learned something during the last two years of education.

On p. 7 the authors argue that “This is because emotional intelligence is not genetically determined…”, even though T. Vernon and colleagues could show in a series of papers that trait EI has a similar genetic influence as other personality traits. In contrast, I guess the authors believe (such as Daniel Goleman in 1995) 80 % of lifetime success is determined by EI (which can be acquired – more or less – by each individual).

Almost thirty years of empiricalresearch could not provide any empirical evidence at all for that inappropriate statement. 

Author Response

Dear reviewer,

Thank you for allowing us to submit our revised manuscript titled: “Emotional intelligence among nursing students: findings from a longitudinal study”. We appreciate the time and effort to providing feedback on our manuscript. We have prepared the revised manuscript according to the comments. In the paragraphs below, we list the comment, followed by our responses. The changes in the revised manuscript are marked with track changes.

We look forward to hearing from you regarding our submission and to respond to any further questions and comments you may have.

Best regards.
Authors

The present study aimed at examining whether EI in nursing students changes over time (from the first to the last year of their education). Moreover, sex and age differences in the two applied self-report EI measures were assessed. After having read this MS carefully I have to make several critical comments outlined in the following.

Dear reviewer, we would like to thank you very much for your detailed review and all the comments you have made, which will contribute to improving the quality of our paper.

To begin with, it is at best a bad joke to examine sex differences (for a high-impact journal such as Healthcare) by using a sample of “6” male students. That is an absolute nonsense and has to be cancelled.

Thank you very much for your comment. We agree. Unfortunately, the representation among nursing students is still unevenly distributed by gender, which is why there is such a discrepancy here. We have removed the part of the results that refer to gender differences and then mentioned this in the limitations.

Although the authors only found a significant difference in self-estimated EI scores between the first and the last year of education for the TEIQue-SF but not for the SSEIT, the authors argued/concluded that these findings indicate that EI skills can be learned (without discussing/interpreting the different findings for the two EI questionnaire measures with any single word).

Some authors e.g. Snowden et al. 2015, 2017 suggest the difference could be due to the type of EI being measured. TEIQ is designed to measure EI as a trait, whereas SSEIT sees EI more as an ‘ability’, in other words, more stable over time. 

Second and even more important, it remains unclear whether the EI differences are due to differences in age, nursing education, or social desirability and so on. From my point of view the authors should also have administered a performance-based measures of emotional abilities/skills instead of using two self-estimate questionnaires (by the way, the authors did not provide a rational for using two EI questionnaires). From my point of view, it is not convincing to assess whether students think/believe that they have learned something during the last two years of education.

Thank you for your comment. We agree and have added text in the Limitation section where we emphasize that further studies should use performance-based measures of EI and not only instruments that report students’ own level of EI. We have added also a reference supporting this.

On p. 7 the authors argue that “This is because emotional intelligence is not genetically determined…”, even though T. Vernon and colleagues could show in a series of papers that trait EI has a similar genetic influence as other personality traits. In contrast, I guess the authors believe (such as Daniel Goleman in 1995) 80 % of lifetime success is determined by EI (which can be acquired – more or less – by each individual). Almost thirty years of empiricalresearch could not provide any empirical evidence at all for that inappropriate statement. 

Thank you for your comment and the information you provided. We have removed the part about EI not being genetically determined. However, we still agree with the points made by many authors that EI also develops over the years and through the life experiences of everyone, so we have left this part of the argument.

Reviewer 2 Report

The paper focuses on an interesting theme:  the emotional intelligence among nursing students. This is a longitudinal study, which is undoubtedly an added value of this work. The theme is well-founded through a literature review that, despite being very brief, is consistent.

The objectives are correctly defined. The methodology is well-designed and is consistent with the objectives of the study. The interpretation and discussion of results is clear, objective, and consistent. The conclusions summarize well the results obtained and are consistent with the work presented.

The greatest weakness of this work lies in the statistical analysis, which is very poor.

Although the study is supposed to be longitudinal, the authors limit themselves to comparing the IE scores in the first and third years. Comparisons of scores according to gender and age must be carried out separately for the first year and for the third year. In addition to being more consistent, this analysis will enrich the interpretation and discussion of results.

It would also be very interesting to enrich this analysis considering the remaining socio-demographic variables.

Author Response

Dear reviewer,

Thank you for allowing us to submit our revised manuscript titled: “Emotional intelligence among nursing students: findings from a longitudinal study”. We appreciate the time and effort to providing feedback on our manuscript. We have prepared the revised manuscript according to the comments. In the paragraphs below, we list the comment, followed by our responses. The changes in the revised manuscript are marked with track changes.

We look forward to hearing from you regarding our submission and to respond to any further questions and comments you may have.

Best regards.
Authors

The paper focuses on an interesting theme:  the emotional intelligence among nursing students. This is a longitudinal study, which is undoubtedly an added value of this work. The theme is well-founded through a literature review that, despite being very brief, is consistent. The objectives are correctly defined. The methodology is well-designed and is consistent with the objectives of the study. The interpretation and discussion of results is clear, objective, and consistent. The conclusions summarize well the results obtained and are consistent with the work presented.

Dear reviewer, thank you very much for your comments.

The greatest weakness of this work lies in the statistical analysis, which is very poor. Although the study is supposed to be longitudinal, the authors limit themselves to comparing the IE scores in the first and third years. Comparisons of scores according to gender and age must be carried out separately for the first year and for the third year. In addition to being more consistent, this analysis will enrich the interpretation and discussion of results.

It would also be very interesting to enrich this analysis considering the remaining socio-demographic variables.

Thank you very much for your suggestions and comments. We have added to the results data on the differences between the average in the first and third year of study according to the different variables that were part of the questionnaire. We have also then address results in discussion.

Reviewer 3 Report

Dear authors

My congratulations for a good job. The study adds to the body of knowledge about EI in nursing students, especially in the Slovenian context, which, together with the correctness of the analyses, the great robustness of the sample in a longitudinal study that is not often found in research, and structural quality of the article in general, although it is not a great contribution, it is of interest that it be available for future research.

Best regards

Author Response

Dear reviewer,
Thank you for allowing us to submit our revised manuscript titled: “Emotional intelligence among nursing students: findings from a longitudinal study”. We appreciate the time and effort to providing feedback on our manuscript. We have prepared the revised manuscript according to the comments. In the paragraphs below, we list the comment, followed by our responses. The changes in the revised manuscript are marked with track changes.

We look forward to hearing from you regarding our submission and to respond to any further questions and comments you may have.

Best regards,
Authors

Dear authors, my congratulations for a good job. The study adds to the body of knowledge about EI in nursing students, especially in the Slovenian context, which, together with the correctness of the analyses, the great robustness of the sample in a longitudinal study that is not often found in research, and structural quality of the article in general, although it is not a great contribution, it is of interest that it be available for future research. Best regards

Dear reviewer, thank you very much for your feedback. We also believe that this paper represents the beginning of research in this field in the Slovenian environment. Best regards

Round 2

Reviewer 1 Report

From my point of view, you have primarily shortened your MS. Yet, several main shortcomings were not be remedied. Some of them are - unfortunately not changeable at all (e.g., missing instruments and measuring data and so on); other objections/comments (which are principally changeable) have not been considered at all.  

Author Response

Thank you very much for your review and for your opinion. We also appreciate all the suggestions made in the first review. We have shortened the paper to remove results that were not relevant due to the unequal gender distribution, as you noted too. However, we have added more results in the form of tables. To improve these results, we have now also added p-values for all variables to the table and further expanded the discussion. We have considered all the suggestions that can now be taken at this stage of the analysis of the results.

A rational for using two EI questionnaires is provided in Measures section. Our interpretation of findings that emotional skills can be learned is revised into could be improved. Vernon et al study was also included emphasising that EI is largely attributable to genetic and nonshared environmental factors.

Unfortunately, as this study is now finished, it is not possible to address all suggestions in the paper. Your suggestions on using performance-based measures of emotional abilities/skills is addressed in Limitation sections and will certainly serve us well when we repeat or continue this study with other students. This article mainly presents a description of the current situation and changes in EI in students over the course of their education and, due to the results obtained, shows the potential for further research in this area, which we may extend based on your suggestions.

Reviewer 2 Report

This is a second review. The authors took into account the reviewers' considerations by making relevant changes that allowed to improve the quality of the manuscript. Nevertheless, it would still be interesting and useful to complete the information contained in Table 2 with the p-values ​​resulting from the comparisons of the mean values ​​for the various domains under study at the two moments under analysis, then proceeding to a more complete interpretation of the results obtained.

Author Response

Thank you for your second review and your suggestion. We appreciate the time and effort reviewers have dedicated to providing feedback on our manuscript. We have extended the results table further and added a p-value for both tools for all variables in the first and third year. We have also expanded the discussion based on the additional results.